# The Ruminal Microbiome and Metabolome Alterations Associated with Diet-Induced Milk Fat Depression in Dairy Cows

**DOI:** 10.3390/metabo9070154

**Published:** 2019-07-23

**Authors:** Hongbo Zeng, Changzheng Guo, Daming Sun, Hossam-eldin Seddik, Shengyong Mao

**Affiliations:** 1Laboratory of Gastrointestinal Microbiology, College of Animal Science and Technology, Nanjing Agricultural University, Nanjing 210095, China; 2Jiangsu Key Laboratory of Gastrointestinal Nutrition and Animal Health, Nanjing Agricultural University, Nanjing 210095, China; 3National Experimental Teaching Demonstration Center of Animal Science, Nanjing Agricultural University, Nanjing 210095, China; 4National Center for International Research on Animal Gut Nutrition, Nanjing Agricultural University, Nanjing 210095, China; 5Joint International Research Laboratory of Animal Health and Food Safety, Nanjing Agricultural University, Nanjing 210095, China

**Keywords:** milk fat depression, metabolome, microbiome, rumen

## Abstract

Milk fat depression (MFD) syndrome represents a significant drawback to the dairy industry. The aim of this study was to unravel the ruminal metabolome-microbiome interaction in response to diet-induced MFD in dairy cows. Twelve healthy second parity Holstein dairy cows (days in milk (DIM) = 119 ± 14) were randomly assigned into control (CON, *n* = 6) group and treatment (TR, *n* = 6) group. Cows in TR group received a high-starch total mixed ration (TMR) designed to induce an MFD syndrome. Decreased milk fat yield and concentration in TR cows displayed the successful development of MFD syndrome. TR diet increased the relative abundance of *Prevotella* and decreased the relative abundance of unclassified *Lachnospiraceae*, *Oribacterium*, unclassified *Veillonellaceae* and *Pseudobutyrivibrio* in ruminal fluid. Metabolomics analysis revealed that the ruminal fluid content of glucose, amino acids and amines were significantly increased in TR cows compared with CON cows. Correlation analysis revealed that the concentration of amines and amino acids were highly correlated with the abundance of *Oribacterium*, *Pseudobutyrivibrio*, *RC9_gut_group*, unclassified *BS11_gut_group* and *Selenomonas*. In general, these findings revealed that TR diet reduced the rumination time and altered rumen fermentation type, which led to changes in the composition of ruminal microbiota and metabolites, and caused MFD.

## 1. Introduction

Milk fat depression (MFD), defined as the reduction in the yield and percentage of milk fat, is a serious problem in dairy industry [1]. MFD is a classic example of the interactions among dietary nutrients, the gastrointestinal microbiota and tissue physiology [2]. There have been various proposed theories to explain the occurrence of MFD in dairy industry, and the most widely accepted theory was that under the condition of large-scale dairy farm, dairy cows fed high concentrate diets had a decrease of milk fat because of the changes in the concentration of volatile fatty acids (VFA) and the alteration of microbiota in the rumen [3]. Urrutia et al. [4] reported that both the butyrate and acetate concentrations and ratio of acetate to propionate in the rumen are related to the milk fat synthesis. The accumulation of VFA is closely related to microbial fluctuations, thus the changes in VFA in the case of MFD indicated an association between the ruminal microbiota and MFD, previous studies showed that the MFD was related to a decrease of Bacteroidetes and an increase of Firmicutes and Actinobacteria. While the unclassified *Lachnospiraceae*, *Butyrivibrio*, *Bulleidia* and *Coriobacteriaceae* bacteria taxa were higher in MFD cattle [3]. Weimer, Stevenson and Mertens [1] reported a shift in ruminal bacterial community composition in lactating dairy cows encountering MFD. These studies proposed the ruminal bacterial community composition can be altered in MFD cattle, however, the relationships between ruminal metabolites and microbiota are not clearly elucidated. Numerous evidences indicated that rumen metabolic disorders related with changes in the ruminal microbiota are important factors for physical disorders in dairy cows [5,6]. Therefore, we need to explore the composition and alterations of ruminal metabolites in MFD cattle which can reveal microbial mediated metabolic processes. Furthermore, it can provide comprehensive associations between the ruminal microbiome and metabolome under conditions of diet-induced MFD.

In the present study, we hypothesized that there is a shift in the global profile of ruminal microbiome and metabolome in dairy cows with MFD induced by a pelleted high-starch diet. Thus, the objective of our research is to establish a model of dairy cows encountering MFD and investigate the associations between ruminal microbiome and metabolome.

## 2. Results

### 2.1. Dry Matter Intake (DMI), Milk Yield and Composition

There was no significance in DMI and milk yield between both groups (Table 1). However, compared with CON group, the milk fat percentage and milk fat yield were lower (*P* = 0.001, *P* < 0.001), while the milk lactose percentage was higher (*P* = 0.028) in TR group. No significant differences were observed in milk lactose, protein percentage and yield between CON and TR groups.

### 2.2. Fermentation Parameters and Animal Behavior

The concentration of propionate and propionate percentage were higher (*P* = 0.002, *P* < 0.001) in TR group compared with CON group (Table 2). There was no significance in lactic acid concentration and total VFA concentration between both groups. The ruminal pH, acetate percentage, butyrate percentage and acetate to propionate ratio were lower (*P* = 0.011, *P* = 0.005, *P* = 0.018, *P* = 0.001) in TR group compared with CON group.

Rumination time was lower (*P* = 0.027) in TR group than that in CON group (Table 3). No significant difference was observed in eating and lying time between CON and TR groups during the monitoring period.

### 2.3. Diversity, Richness and Composition of the Ruminal Bacterial Communities

The number of operational taxonomic units (OTUs), Chao 1 value, Shannon index and abundance-based coverage estimator (ACE) were lower (*P* = 0.001, *P* = 0.001, *P* = 0.002, *P* = 0.002) in TR group than CON group (Table 4). The results of Principal coordinates analysis (PCoA) profile revealed that the plots in CON and TR groups were definitely detached (Figure 1; axis 1 + axis 2 = 89.11%).

We identified 22 bacterial phyla in ruminal fluid samples (Table 5). The common of the sequences obtained belonged to Bacteroidetes and Firmicutes. TR group had a higher (*P* = 0.018) relative abundance of Bacteroidetes, with a lower (*P* = 0.028, *P* = 0.006, *P* = 0.045) relative abundance of Firmicutes, Proteobacteria and Candidate_division_TM7 than CON group.

We found 178 taxa across ruminal fluid samples. The abundant taxa (relative abundance of ≥ 0.01%) were presented in Table 6 for the purposes of clarity and visualization. The *Prevotella* taxa was higher (*P* = 0.018), while the unclassified *Lachnospiraceae*, *Oribacterium*, unclassified *Christensenellaceae*, *Pseudobutyrivibrio*, unclassified *Veillonellaceae* and unclassified *Succinivibrionaceae* taxa were lower (*P* = 0.045, *P* = 0.010, *P* = 0.044, *P* = 0.010, *P* = 0.011, *P* = 0.028) in TR group than CON group. No significance in the abundance of *Succiniclasticum*, *Roseburia*, *Ruminococcus* et al. between CON and TR groups.

### 2.4. Predicted Functions of Ruminal Bacterial Microbiota

This study inferred that 43 gene families were identified in the rumen fluid samples. The gene families of energy metabolism, carbohydrate metabolism, lipid metabolism, amino acids metabolism, signal transduction and translation were the most abundant gene families in the rumen microbiome of both groups (Figure 2). Furthermore, it was noticed that three gene families in the ruminal microbiome showed a significantly different abundance between both groups. When compared with CON group, the gene families involved in the carbohydrate metabolism, energy metabolism and transcription were significantly increased (*P* = 0.021, *P* = 0.030, *P* = 0.004) in TR group.

### 2.5. Correlation Analysis between the Ruminal VFA Parameters and Microbiome

Correlation networks were constructed using the data regarding the VFA parameters and the most abundant taxa (relative abundance of ≥ 0.01%) based on Spearman correlation coefficients (|*r*| > 0.60 and *P* < 0.05) in CON and TR groups (Figure 3). We observed that the concentration of acetate (*P* = 0.003, *r* = −0.809) and butyrate (*P* = 0.025, *r* = −0.682), the ratio of acetate to butyrate (*P* = 0.011, *r* = −0.745) had negative correlation with the *Prevotella* taxa. Acetate concentration (*P* = 0.011 and 0.011, *r* = 0.745 and 0.745) and the ratio of acetate to propionate (*P* = 0.022 and 0.047, *r* = 0.690 and 0.618) were positively correlated with the Unclassified *Ruminococcaceae* and Unclassified *Lachnospiraceae* taxa. We also found that propionate had not high correlation with any other ruminal bacteria. Furthermore, the ratio of acetate to propionate was sensitive to the *Prevotella* taxa from its correlation value. The *Pseudobutyrivibrio* taxa had positive correlation with acetate (*P* = 0.003, *r* = 0.808) and butyrate (*P* < 0.001, *r* = 0.867) concentration, but it had no correlation with the ratio of acetate to propionate.

### 2.6. GC/MS Analysis of the Ruminal Fluid

We detected 665 valid peaks unique in the ruminal fluid sample. We identified 272 metabolites, mainly amines, amino acids, organic acids, fatty acids, lipids, sugars, nucleosides and other metabolites. The metabolites of principal component analysis (PCA) and partial least squares-discriminate analysis (PLS-DA) in CON and TR groups were conducted. As shown in Figure 4, the PCA discovered that the axes 1 and 2 were up to 31.4% and 14.8% of the total variation, respectively. The ruminal metabolites in CON and TR groups were clearly distinct, demonstrating the difference of metabolites in the rumen between CON and TR groups.

### 2.7. Differences in Ruminal Metabolites between CON and TR Groups

In total, 29 differential metabolites (DMs) (false discovery rate (FDR) < 0.05 and variable importance in projection (VIP) > 1) were identified according to the statistical analysis and the VIP value obtained from the PLS-DA analysis. These DMs were classified and the name, the VIP value and the fold change of each metabolite were shown in Table 7. In general, the main difference in ruminal metabolites between CON and TR groups were the variation of fatty acids, amino acids, lipids, organic acids, sugars, amines, and nucleosides. The content of 23 ruminal metabolites, such as amino acids (glutamine, isoleucine, ornithine, oxyproline, alanine and serine), sugars (6-deoxy-d-glucose and d-talose), amines (*n*-Acetyl-beta-d-mannosamine, malonamide, 5-methoxytryptamine and indole-3-acetamide), methyl phosphate, pimelic acid, taxifolin and malonate in TR group were higher (FDR < 0.05) than CON group.

### 2.8. Metabolic Pathways of Differential Metabolites

To provide a comprehensive view of the DMs between dairy cows with MFD and the controls, pathway analysis was conducted (Figure 5). Results revealed that glycine, serine and threonine metabolism; biosynthesis of amino-acetyl-tRNA; methane metabolism; amino acids and nucleoside metabolism; arginine and alanine metabolism; glutamate metabolism were significantly enriched (FDR < 0.05) in TR group compared with CON group.

### 2.9. Correlation Analysis between the Ruminal Microbiome and Metabolome

Correlation networks were constructed by means of the data concerning the most abundant taxa (relative abundance of ≥ 0.01%) and DMs (VIP >1 and FDR < 0.05) created on Spearman correlation coefficients (|r| > 0.75 and *P* < 0.05) in CON and TR groups (Figure 6). We observed that the top 5 high negative correlations were between the unclassified *BS11_gut_group* and serine; the *RC9_gut_group* and oxyproline; *Oribacterium* and indole-3-acetamide; the *RC9_gut_group* and 5-methoxytryptamine; the unclassified *Lachnospiraceae* and glutaconic acid. The top 5 high positive correlations were between *Treponema* and d-talose; the unclassified *Veillonellaceae* and 3-hydroxyphenylacetic acid; the *Prevotella* and methyl phosphate; the *Prevotella* and succinic acid; the *Prevotella* and glutaconic acid. The main bacteria were *Prevotella*, *Syntrophococcus*, unclassified *Veillonellaceae Oribacterium*, unclassified *Lachnospiraceae*, *RC9_gut_group* and unclassified *BS11_gut_group* which belong to Bacteroidetes and Firmicutes; the main metabolites were amines (indole-3-acetamide and 5-methoxytryptamine), amino acids (serine and oxoproline), dicarboxylic acid (glutaconic acid), organic acid (succinic acid) and sugars (d-talose). In addition, alanine was negatively (*r* = −0.866; *P* = 0.021) correlated with the *Oribacterium*.

## 3. Discussion

Fat is the most variable of the major components of ruminant milk, which can be affected by diet and the proportion of ruminal VFA [7]. Previous studies have revealed that there was an association between shifts in the rumen VFA patterns and the decrease in yield of milk fat [7,8]. Coch et al. [8] observed that diet-induced MFD cattle was related to the shifts in molar proportions of ruminal VFA. In consistency with these reports, in the present study, the cows in TR group encountered a decrease in percentage of acetate and butyrate and an increase in propionate percentage compared with CON group. Thus, the main reason for the MFD occurrence could be the decrease in percentage of butyrate and acetate.

Saliva production is an important characteristic of the ruminant digestive process because saliva acts as a buffering agent that helps to preserve optimum pH levels in the rumen. Saliva secretion occurs continuously but is more secreted during eating and ruminating [9]. Rumination is a natural behavior of foregut fermenting species and it can stimulate the saliva secretion which maintains an optimum rumen environment. By decreasing the forage particle size through pelleting [10], various studies have determined pelleted diets’ possessions on the feeding behavior of dairy cows. In this study, compared with TMR, pelleted diet altered the feeding behavior of dairy cows because of its own characteristics, and we found that the rumination time of dairy cows in TR group was decreased compared with CON group, and this was caused by short forage particle size of TR diet. TR diet had a higher proportion of maize with additional higher proportion of corn germ meal and corn husk, which provided large amounts of readily digestible carbohydrates. Mechanical activation and high temperature would change the physical structure of the starch and could enhance the starch degradability in the rumen [11]. In the present study, the ingredients of the TR diet were chopped, ground and pelleted through high exit temperature (84 °C) by the pelleted machine. This result indicated that TR diet could enhance the degradability of starch in the rumen and caused MFD in dairy cows. Expectedly, cows in the TR group had a lower time of rumination, which may cause the lower ruminal pH which was similar to the previous studies [12,13]. These results indicated that the time of rumination in dairy cows can partially change the ruminal pH by means of saliva secretion.

The richness and diversity of ruminal bacterial microbiota are important indicators of their normal physiological state. In the present study, the lower Chao 1 and Shannon index in TR group suggested that the microbial community composition was altered and tended to be less diverse of dairy cows in TR group. Belanche et al. [14] found that compared with fiber rich diet, starch rich diet can decrease the bacterial and fungal diversity in the rumen. The reason was certain microorganisms were sensitive to nitrogen and rumen ammonia concentration. The accumulation of amines could exert negative effect on the ruminal microbiota. In our study, the metabolites of certain amines were increased in TR group which was consistent with Belanche et al. [14]. We also found that the relative abundance of Bacteroidetes was higher and the Firmicutes and Proteobacteria were lower in TR group in the present study. A replacement of Firmicutes and Proteobacteria by Bacteroidetes was associated with the transition from the forage to concentrate diets [15]. This indicated the Bacteroidetes became the dominant bacteria by the high starch diet. Our results in the present study confirmed the previous reports that *Prevotella* was the most abundant genus of adult dairy cows [16]. Reduced eating time with increased starch content in diets could result in the high relative abundance of *Prevotella*, which could be explained by the increased proportion of ruminal fermentable substrates and the reduction in ruminal pH [17]. Correlation analysis revealed the relative abundance of *Prevotella* was negatively correlated with the acetate, butyrate concentration and the ratio of acetate to propionate in the present study, which indicated that high relative abundance of *Prevotella* changed the ruminal fermentation type. The main reason was that TR diet provided larger proportion of ruminal fermentable substrates, changed the ruminal pH and caused the shift of the ruminal bacterial community composition, which altered the VFA production. In this study, high-starch diet provided large amounts of readily digestible carbohydrates which could increase the relative abundance of *Prevotella*, and the *Prevotella* taxa would participate in the carbohydrates digesting process. Meanwhile, the lower ruminal pH could provide the suitable conditions for *Prevotella* to survive, which was consistent with James [18]. Consistent with the findings by Liu et al. [19], some higher abundance of unclassified groups, containing unclassified *Lachnospiraceae* and unclassified *Ruminococcaceae* were observed in rumen of dairy cattle. Kim et al. [20] found that these two unclassified bacterial groups might play a significant role in fiber digestion of rumen. In the present study, the dairy cows in TR group had lower relative abundance of unclassified *Lachnospiraceae* and unclassified *Ruminococcaceae*, correlation analysis revealed that unclassified *Lachnospiraceae* and unclassified *Ruminococcaceae* had positive correlation with the acetate concentration and the ratio of acetate to propionate, which indicated that both of the bacteria could be intensively concerned with high starch diets fermentation in the rumen. The ruminal microbiome provides many physiological functions that are required by the host dairy cow. To define the potential functions of the microbiota in ruminal fluid samples, the PICRUSt was used to infer putative metagenomes from the 16S rRNA gene profiles [21]. Results showed that the ruminal microbiota had an enhanced capacity to affect carbohydrate metabolism and energy metabolism, while it had a decreased capacity of transcription and aging of dairy cows in TR group. These results implied that rumen bacteria and its functions could respond quickly to high levels of starch in TR diets, and adapt to the diets by enhancing the carbohydrate and energy metabolism.

Metabolomics analysis revealed that the levels of glutamine, isoleucine, ornithine, oxyproline, alanine and serine were higher in dairy cows of TR group. Trent et al. [22] found that the concentration of alanine was concerned with the demise of Gram-negative and Gram-positive bacteria. In this study, correlation analysis revealed that alanine was negatively correlated with *Oribacterium*. Therefore, the increased concentration of alanine in the rumen of dairy cows with MFD appeared to be caused by the alternation in the abundance of *Oribacterium* in this study. From the catabolic pathway of arginine in many microorganisms [23], ornithine has been revealed to be formed by ornithine carbamoyl-transferase. Correlation analysis revealed that the ornithine had high positive correlation with the *Prevotella* and negative correlation with the *Pseudobutyrivibrio.* In this study, the metabolism of ornithine was higher in TR group. This finding indicated that the genus of *Prevotella* and *Pseudobutyrivibrio* may be correlated with the ornithine production through the catabolic pathway of arginine in dairy cows with MFD. From the present study, the levels of biogenic amines including *n*-acetyl-beta-d-mannosamine, malonamide, 5-methoxytryptamine, indole-3-acetamide were increased in TR group. Ruminants possibly obtained biogenic amines from both diets and microbial metabolites in the rumen and the biogenic amines were mainly instigated from the decarboxylation of convinced amino acids [24]. Moreover, the enriched pathways including biosynthesis of valine, leucine, isoleucine and amino-acetyl-tRNA biosynthesis was also observed in cows of TR group. These findings indicated that ruminal microbial bacteria enhanced amino acids metabolism. The possible explanation for the increase in the amino acid metabolism is supported by the lower ruminal pH, and previous studies revealed that the increased activity of bacterial amino acid decarboxylases was owing to the decreased ruminal pH [25,26].

For dicarboxylic acids, malonate, pimelic acid, azelaic acid and glutaconic acid were identified in this study. The dicarboxylic acids were produced by the biohydrogenation of the non-conjugated linoleic acids formed from linoleic [27,28]. Previous studies suggested that the rate of bio-hydrogenation in the rumen was linked to the milk fat depression (MFD) [29]. Correlation analysis revealed that the glutaconic acid had a higher negative correlation with unclassified *Ruminococcaceae*, unclassified *Lachnospiraceae* and unclassified *Christensenellaceae*, meanwhile, the benzoic acid and 3-hydroxyphenylacetic acid had a higher positive correlation with the unclassified *Veillonellaceae*. These results indicated that these unclassified bacteria such as unclassified *Ruminococcaceae*, unclassified *Lachnospiraceae*, unclassified *Christensenellaceae* and unclassified *Veillonellaceae* may be related with the fatty acids metabolism combined with the rate of bio-hydrogenation. This finding was consistent with Huws [30] who observed that unclassified *Clostridiales* and *Ruminococcaceae* exert significant effect on bio-hydrogenation. However, these data did not conclude that these bacteria are definitively elaborate in bio-hydrogenation. Further studies should aim to investigate the exact functions of these bacteria in bio-hydrogenation.

## 4. Materials and Methods

### 4.1. Experimental Design and Animals

The animal experimental design and procedures of this study were supported by the Animal Care and Use Committee of Nanjing Agricultural University following the requirements of the Regulations for the Administration of Affairs Concerning Experimental Animals (The State Science and Technology Commission of P. R. China, 1988. No. SYXK (Su) 2015-0656). Twelve healthy second parity Holstein dairy cows (days in milk (DIM) = 119 ± 14, BW = 628 ± 56 kg, mean ± SD, milk yield = 30–40 kg/d) were randomly allocated into control (CON, *n* = 6) group and treatment (TR, *n* = 6) groups. Cows were fed three times daily (0600 h, 1300 h and 2000 h) and milk yield and feed intake were recorded every day. Cows were housed in individual tie-stalls fitted with rubber mats and had continuous access to water from water bowls during the experiment. The whole experimental period lasted for 55 days, and all the samples were collected from day 51 to 55 of the experimental period.

### 4.2. Diets and Feeding

The experimental diets were formulated according to the lactation nutritional requests of a 30 kg milk/d producing cow (NRC, 2001). Dairy cows in CON group were fed TMR, while cows in TR group were fed pelleted high-starch TMR diet. We allowed approximately 5% feed residual during the experiment period. The pelleted ration was pelleted with a pelleting machine (Jiangsu Muyang Group Co., Ltd., Jiangsu, China) with steam added at an 84 °C exit temperature. Pellet diameter was 0.63 cm and the length was 1.75 cm. The chemical composition and ingredients of diets are presented in Table 8.

### 4.3. Sampling and Analysis

Feeding behavior analysis. Two monitors (AW-W815ZL4-Y; Guangdong; China) were used for monitoring the behavior (24 h/d) of dairy cows (6 cows each) on the 51st, 52nd and 53rd day of the experimental period. Each monitor’s data recorded the time that cows spent eating, lying or ruminating and then were summed each day.

### 4.4. Feed Sampling and Analysis

Feed and residual samples were collected at 0600 h, 1300 h and 2000 h after each milking on the 51st, 52nd and 53rd day of the experimental period. Furthermore, samples were kept at −20 °C until chemical composition analysis. Crude protein, moisture, starch, ash and ether extract were measured according to the AOAC methods [31]. Acid detergent fiber (ADF) and neutral detergent fiber (NDF) were measured followed by Van Soest et al. [32] (Table 8).

### 4.5. Milk Sampling and Analysis

Milk samples were collected at 0600 h, 1300 h and 2000 h after each milking on the 51st, 52nd and 53rd day of the experimental period. Sections were reserved at 4 °C, using borneol as a conserving and were sent to Shanghai DHI organization (China) for analyzing. The content of fat, protein and lactose of milk were measured by a near-infrared analyzer (Foss Electric, Denmark).

### 4.6. Ruminal Fluid Sampling and Analysis

Ruminal fluid samples were collected through oral stomach tube (OST) [33] for 2 consecutive days at 6 h after morning feeding once per day on the 54th and 55th day of the experimental period. In order to obtain representative ruminal fluid samples, the insertion depth of the OST from the animal’s incisors teeth to central rumen was 2 m according to the method of Shen, Chai, Song, Liu and Wu [33]. The ruminal fluid was collected about 40 mL through spontaneous efflux after aspirating with a modified 60 mL syringe and strained through 4 layer cheesecloth. The ruminal fluid pH was measured directly using a calibrated pH meter (Starter 300; Ohaus Instruments Co. Ltd., Nanjing, China). Samples were kept at −80 °C until analyzed for fermentation parameters, microbial DNA extraction and GC/MS analysis. The rumen filtrate was thawed and centrifuged at 3000× *g* for 10 min. Gas chromatography (7890A, Agilent, Palo Alto, CA, USA) was utilized for VFA analysis followed by Mao et al. [34]. The lactate level was detected by a lactic acid assay kit (Nanjing Bioengineering Institute, China).

### 4.7. Microbial DNA Extraction and High-Throughput Sequencing

Bead-beating approach followed by phenol-chloroform extraction was utilized to ensure an efficient extraction of the total microbial DNA, this protocol had been described by Mao et al. [35]. Nanodrop spectrophotometer (Nyxor Biotech; Paris, France) was used to detect DNA yield and quality. Ultimately, high-quality DNA samples were kept at −80°C for high-throughput sequencing. The V3-V4 region of the 16S ribosomal RNA (rRNA) bacterial genes was amplified using (341F = 5′-CCTAYGGGRBGCASCAG-3′; 806 R = 5′-GGACTACCVGGGTATCTAAT-3′) primers. The barcode was a sample-unique sequence of 6 bases. PCR was conducted based on the Accuprime Taq DNA polymerase System (Invitrogen, Carlsbad, CA), which includes an initial denaturation at 95℃ for 2 min, 30 cycles of 20 s denaturation at 95 °C, 20 s annealing at 60 °C, and 60 s extension at 65 °C, and a 65 °C extension for 7 min [36]. The amplicons from each DNA sample were fluorometrically quantified, normalized, and pooled. Finally, 454 GS FLX Titanium chemistry platform (Majorbio Bio-Pharm Technology, Shanghai, China) was utilized for DNA sequencing.

QIIME software platform (http://qiime.org/) [37] was used for the microbial data analysis. According to the previously assigned sample-specific barcodes, high quality sequences were carefully chosen after decoding. Length of fragments more than 250 bps without unknown bases and average score of quality higher than 25 were taken as criteria to select sequences. OTU picking was performed based on 97% level sequence comparison, using UPARSE [38]. The representative sequence of each OTU was identified as the most abundant sequence in an OTU, and then the most abundant sequences of all OTUs were allocated to the Green genes 13.5 core set [37].

Ribosomal Database Project Bayesian classifier was used to perform the taxonomy assignment [39] via the contrasting of descriptive sequences in each OTU with the Green genes database [40]. QIIME (http://qiime.org/) software package was utilized for evaluating the ACE, Chao1, Shannon, and Simpson. Unweighted Unifrac metric PCoA was measured utilizing MOTHUR (https://www.mothur.org) program [41]. The 16S sequencing data were acquiesced to the Sequence Read Archive (SRA) under consent PRJNA534384.

### 4.8. Predicted Rumen Functions of Bacterial Microbiota

To more deeply understand the rumen functions of bacterial microbiota, we used PICRUSt to predict the metagenomics involvement of the detected communities [21].

### 4.9. Metabolomics Analysis

For metabolite analysis, 200 μL ruminal fluid containing 10 μL of adonitol as an internal standard was extracted with 350 μL methanol and vortexed for 30 s. Samples were centrifuged at 12,000 rpm for 15 min at 4 °C. After centrifugation, 350 μL of the supernatant was evaporated to dryness using vacuum concentrator without heating. After evaporation, 50 μL of methoxide was added into amination hydrochloride followed by 30 min incubation at 80 °C condition. Meanwhile, 70 μL of chlorotrimethylsilane silylating reagent was added into the aliquots after 1.5 h incubation at 70 °C. A gas chromatography system (Agilent 7890, Palo Alto, CA, USA) coupled with a Pegasus HT (LECO, Shanghai, China) time-of-flight mass spectrometer (GC-TOF-MS) was used to identify the metabolites fitted with a DB-5MS capillary column (30 m × 250 μm inner diameter, 0.25 μm film thickness; J&W Scientific, Folsom, CA, USA). 1μL aliquot of the analyte was injected in splitless mode. Helium was used as the carrier gas, the front inlet purge flow was 3 mL min^−1^, and the gas flow rate through the column was 1mL min^−1^. The primary temperature was kept at 50 °C for 1 min, then raised to 310 °C at a rate of 10 °C min^−1^, and finally kept for 8 min at 310 °C. Chroma TOF 4.3X software and LECO-Fiehn Rtx5 database (Shanghai Biotreee biotech Co. Ltd., Shanghai, China) were used for raw peaks exacting, the data baselines filtering and calibration of the baseline, peak alignment, deconvolution analysis, peak identification and integration of the peak area. LECO-Fiehn Rtx-5 database was utilized to identify metabolites.

### 4.10. Statistical Analysis

Data of feeding behavior, milk yield, DMI, milk composition and ruminal fermentation parameters were analyzed through the independent sample t-test procedures (SPSS v. 20, SPSS Inc., Chicago, IL, USA). The non-parametric Kruskal-Wallis test was carried out to test for significant variation in the relative abundance of microbiota and the predicted KEGG pathways (%) (SPSS v. 20, SPSS Inc., Chicago, IL, USA). *P* < 0.05 was defined as statistical significance.

SIMCA-P^+^ 13.0 software (Umea, Sweden) was utilized for conducting multivariate statistical analysis together with PCA. DMs between both groups were identified combining VIP, FDR and FC (TR/CON) acquired from PLS-DA analysis and statistical analysis (VIP > 1, FDR < 0.05 and FC > 1.5 or < 0.67). To ensure data quality, quality control samples were inserted randomly. Data for DMs were then exported into the MetaboAnalyst web server (https://www.metaboanalyst.ca/) to explore its metabolic pathway spreading and enrichment analysis [42]. The FC (TR/CON) was the ratio of mean value of peak area gained from cows in CON and TR groups.

Correlations between ruminal VFA parameters and the relative abundance of microbiota were calculated using Spearman’s correlation test implemented in XLStat (Addinsoft, Paris, France). Heml 1.0.3.3 (Heatmap Illustrator) software was used to visualize the correlation that had a correlation coefficient absolute value greater than 0.60 and significance less than 0.05.

Correlations between metabolites and microbiota were calculated using Spearman’s correlation test implemented in XLStat (Addinsoft, Paris, France). Gephi 0.8.2 software (https://gephi.org/) was used to envision the correlation between microbiota and metabolites that had a correlation coefficient absolute value greater than 0.75 and significance less than 0.05.

## 5. Conclusions

In the present study, under the condition of dairy cows encountering MFD, the richness and diversity of the ruminal microbiota were significantly reduced, and ruminal fermentation and the bacterial community composition were altered. Metabolomics analysis revealed that some ruminal metabolites were significantly changed, which increased glucose, amines, amino acids and short chain fatty acids. Moreover, correlation analysis revealed some associations between the microbial bacteria and the metabolites. Overall, high-starch diet reduced the rumination time and changed the composition of ruminal bacteria and metabolites, altered rumen fermentation type, and caused MFD.

## Figures and Tables

**Figure 1 metabolites-09-00154-f001:**
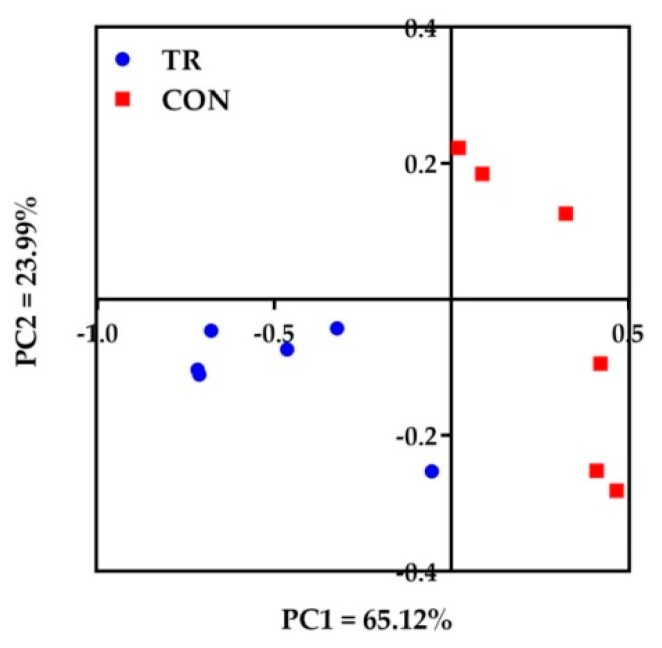
Unweighted UniFrac metric PCoA of microbial diversity in CON and TR groups. The percentage of variation explained by PC1 and PC2 are indicated on the axis.

**Figure 2 metabolites-09-00154-f002:**
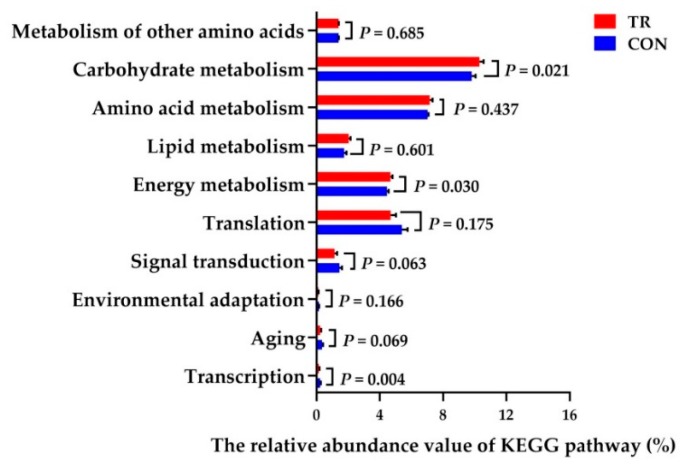
The relative abundance of the top-ten phylogenetic investigation of communities by reconstruction of unobserved states (PICRUSt)-predicted metabolic pathways of ruminal bacterial microbiome in CON and TR groups.

**Figure 3 metabolites-09-00154-f003:**
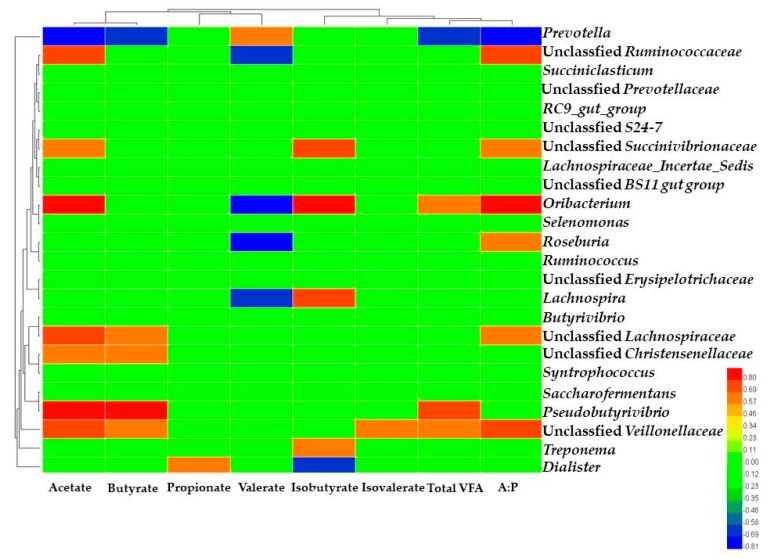
Heat map showing the spearman correlation between ruminal volatile fatty acids (VFA) parameters and microbiome in CON and TR groups. Red means a significant positive correlation, blue means a significant negative correlation and green means a non-significant correlation. The correlation coefficients with a statistical *P* value < 0.05 and the absolute value > 0.60 were used to establish the network graph. A: P = the ratio of acetate to propionate; VFA = volatile fatty acid.

**Figure 4 metabolites-09-00154-f004:**
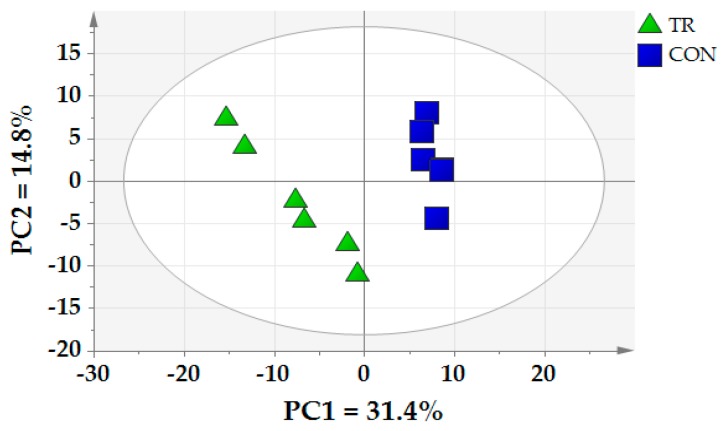
Principal component analysis (PCA) of ruminal metabolites of CON and TR groups. PC1 = the first principal component; PC2 = the second principal component.

**Figure 5 metabolites-09-00154-f005:**
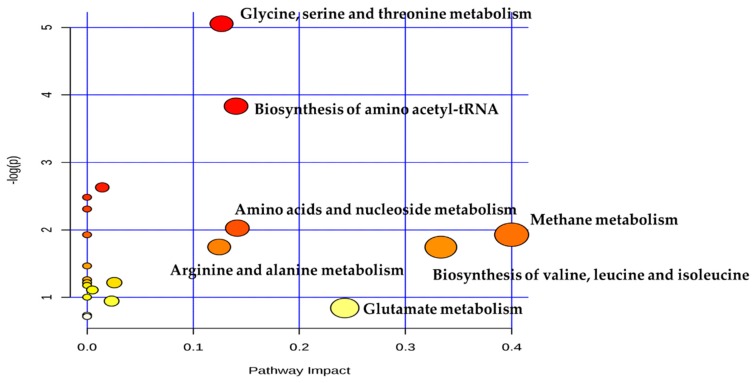
Metabolome view map of the differential metabolites (DMs) (VIP > 1, FDR < 0.05) identified in CON and TR groups. The larger size indicates higher pathway enrichment, and darker color indicates higher pathway impact values.

**Figure 6 metabolites-09-00154-f006:**
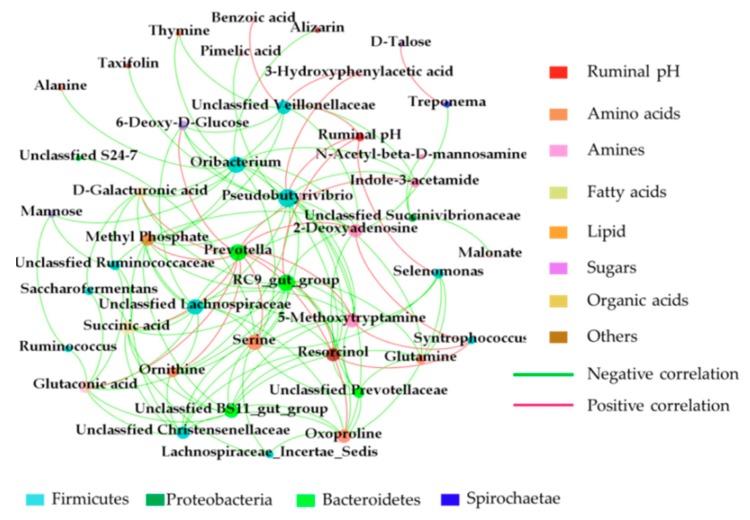
The spearman correlation network between ruminal microbiome and metabolome in CON and TR groups. The correlation coefficients with a statistical *P* value < 0.05 and the absolute value > 0.75 were used to build the network graph.

**Table 1 metabolites-09-00154-t001:** Dry matter intake and milk component yields and concentrations of control (CON ^1^) and treatment (TR ^2^) groups.

Item	CON	TR	SEM	*P*-Value
DMI (Kg/d)	23.72	20.82	0.68	0.639
Milk yield (Kg/d)	35.72	31.36	1.41	0.136
Protein yield (Kg/d)	1.10	0.99	0.04	0.172
Lactose yield (Kg/d)	1.78	1.60	0.07	0.182
Fat yield, (Kg/d)	1.24	0.77	0.04	<0.001
Protein (%)	3.10	3.15	0.02	0.067
Lactose (%)	5.00	5.11	0.02	0.028
Fat (%)	3.47	2.42	0.09	0.001

^1^ CON (control group) = dairy cows fed total mixed ration (TMR). ^2^ TR (treatment group) = dairy cows fed pelleted high-starch diet.

**Table 2 metabolites-09-00154-t002:** Ruminal fermentation parameters of CON ^1^ and TR ^2^ groups.

Item	CON	TR	SEM	*P*-Value
pH	6.32	6.04	0.06	0.011
Lactate (mmol/L)	0.22	0.28	0.05	0.524
Acetate (mmol/L)	69.39	63.11	1.90	0.104
Propionate (mmol/L)	30.14	35.33	1.02	0.002
Isobutyrate (mmol/L)	0.63	0.27	0.11	0.072
Butyrate (mmol/L)	13.61	11.15	0.66	0.074
Isovalerate (mmol/L)	1.55	1.04	0.10	0.011
Valerate (mmol/L)	1.84	2.73	0.17	0.007
Total VFA ^3^ (mmol/L)	117.16	113.63	2.75	0.520
A: P ^4^	2.33	1.79	0.09	0.001
Acetate (%)	59.21	55.46	0.68	0.005
Propionate (%)	25.76	31.25	0.81	<0.001
Butyrate (%)	11.58	9.68	0.40	0.018

^1^ CON (control group) = dairy cows fed total mixed ration (TMR). ^2^ TR (treatment group) = dairy cows fed pelleted high-starch diet. ^3^ VFA = volatile fatty acid. ^4^ A: P = the ratio of acetate to propionate.

**Table 3 metabolites-09-00154-t003:** Average durations of daily feeding behavior (in minutes) observed in CON ^1^ and TR ^2^ groups.

Item	CON	TR	SEM	*P*-Value
Eating	300.19	278.28	3.56	0.201
Lying	860.32	830.73	2.78	0.108
Rumination	380.51	305.65	3.13	0.027

^1^ CON (control group) = dairy cows fed total mixed ration (TMR). ^2^ TR (treatment group) = dairy cows fed pelleted high-starch diet.

**Table 4 metabolites-09-00154-t004:** Alpha diversity of ruminal bacterial communities of CON ^1^ and TR ^2^ groups.

Item	CON	TR	SEM	*P*-Value
Number of OTUs ^3^	1132	544	109	0.001
ACE ^4^	1404	813	112	0.002
Chao 1 value	1416	760	124	0.001
Shannon index	4.92	3.05	0.36	0.002

^1^ CON (control group) = dairy cows fed total mixed ration (TMR). ^2^ TR (treatment group) = dairy cows fed pelleted high-starch diet. ^3^ OTUs = operational taxonomic units. ^4^ ACE = abundance-based coverage estimator.

**Table 5 metabolites-09-00154-t005:** The relative abundance of phylum level (% of total sequences) in ruminal bacterial communities of CON ^1^ and TR ^2^ groups.

Phylum	Relative Abundance (%)		SEM	*P*-Value
	CON	TR		
Bacteroidetes	54.80	79.88	5.95	0.018
Firmicutes	36.81	18.45	5.33	0.028
Proteobacteria	5.99	0.60	1.67	0.006
Spirochaetae	0.56	0.30	0.15	0.522
Cyanobacteria	0.48	0.26	0.06	0.068
Actinobacteria	0.19	0.19	0.03	0.855
Candidate_division_TM7	0.26	0.09	0.04	0.045
Tenericutes	0.38	0.09	0.08	0.067
Fibrobacteres	0.19	0.08	0.05	0.200
Synergistetes	0.06	0.03	0.01	0.234
Candidate_division_SR1	0.17	0.03	0.04	0.120
Planctomycetes	0.01	0.01	<0.01	0.571
Gemmatimonadetes	0.03	<0.01	0.01	0.107
Acidobacteria	0.01	<0.01	0.01	0.284
Others	0.08	<0.01	0.02	0.090

^1^ CON (control group) = dairy cows fed total mixed ration (TMR). ^2^ TR (treatment group) = dairy cows fed pelleted high-starch diet.

**Table 6 metabolites-09-00154-t006:** The relative abundance of ruminal microbiome predominant genera^1^ of CON ^2^ and TR ^3^ groups.

Phylum	Genus	Relative Abundance (%)		SEM	*P*-Value
		CON	TR		
Bacteroidetes	*Prevotella*	42.04	72.91	7.12	0.018
	Unclassified *Prevotellaceae*	3.10	1.66	0.40	0.144
	Unclassified *S24-7*	4.88	3.06	1.01	0.273
	*RC9_gut_group*	2.26	1.02	0.47	0.068
	Unclassified *BS11 gut group*	1.42	0.61	0.42	0.100
Firmicutes	Unclassified *Ruminococcaceae*	7.75	2.28	1.67	0.068
	*Ruminococcus*	3.61	1.83	0.78	0.361
	*Saccharofermentans*	0.66	0.36	0.16	0.201
	*Roseburia*	3.99	1.41	1.34	0.100
	*Butyrivibrio*	1.99	2.25	0.43	0.855
	Unclassified *Lachnospiraceae*	1.62	0.68	0.27	0.045
	*Lachnospiraceae_Incertae_Sedis*	1.53	0.93	0.23	0.201
	*Oribacterium*	1.00	0.10	0.20	0.010
	*Syntrophococcus*	0.86	2.22	0.38	0.144
	*Lachnospira*	0.79	0.36	0.19	0.201
	*Pseudobutyrivibrio*	0.61	0.05	0.12	0.010
	*Selenomonas*	0.97	0.40	0.21	0.100
	*Dialister*	0.54	0.34	0.14	0.855
	Unclassified *Veillonellaceae*	0.54	0.04	0.13	0.011
	Unclassified *Christensenellaceae*	0.99	0.19	0.26	0.044
	*Succiniclasticum*	5.64	3.51	1.05	0.201
	Unclassified *Erysipelotrichaceae*	0.84	0.34	0.20	0.100
Proteobacteria	Unclassified *Succinivibrionaceae*	4.77	0.18	1.58	0.028
Spirochaetae	*Treponema*	0.54	0.29	0.15	0.584

^1^ Taxa are illustrated in alphabetical order. A taxon is assigned as predominant if its relative abundance is ≥ 1% in at least one group. ^2^ CON (control group) = dairy cows fed total mixed ration (TMR). ^3^ TR (treatment group) = dairy cows fed pelleted high-starch diet.

**Table 7 metabolites-09-00154-t007:** List of ruminal fluid metabolites that showed significant difference between CON ^1^ and TR ^2^ groups ^3^.

Compounds	RT ^4^	Mass	VIP ^5^	FDR ^6^	*P*-Value	FC ^7^
Amino acids						
Glutamine	13.75	125	1.69	0.035	0.010	1.85
Isoleucine	10.55	158	1.535	0.043	0.026	2.148
Ornithine	16.91	174	1.558	0.046	0.041	1.948
Oxyproline	13.61	156	1.507	0.044	0.030	3.209
Alanine	7.86	116	1.672	0.046	0.042	4.227
Serine	11.44	204	1.441	0.039	0.016	3.133
Fatty acids						
Malonate	13.24	81	1.776	0.010	<0.001	5.93
Benzoic acid	10.15	179	1.519	0.043	0.028	0.557
Pimelic acid	14.58	125	2.125	0.010	<0.001	2.903
Glutaconic acid	12.6	55	1.255	0.045	0.034	0.345
3-Hydroxyphenylacetic acid	14.65	164	1.788	0.043	0.020	0.44
Azelaic acid	16.72	317	1.532	0.043	0.028	0.477
Lipids						
Methyl Palmitoleate	17.98	97	1.498	0.046	0.044	0.607
Methyl Phosphate	9.14	241	1.54	0.034	0.007	12.668
Sugars						
6-Deoxy-d-glucose	15.74	117	1.838	0.015	0.002	3.49
Mannose	17.53	160	1.089	0.046	0.046	0.588
d-Talose	17.84	299	1.268	0.045	0.034	4.233
Organic acids						
d-galacturonic acid	18.11	333	1.408	0.046	0.046	3.234
Salicylic acid	13.46	267	2.165	0.010	<0.001	0.063
Succinic acid	10.88	147	1.457	0.034	0.007	2.066
Amines						
*N*-Acetyl-beta-d-mannosamine	19.4	73	1.464	0.043	0.026	1.809
Malonamide	13.93	174	1.401	0.043	0.022	2.034
5-Methoxytryptamine	22.8	174	1.714	0.045	0.036	2.049
Indole-3-acetamide	18.53	318	1.863	0.035	0.011	19.487
Nucleosides						
Thymine	12.04	255	1.923	0.046	0.039	3.574
2′-Deoxyadenosine	24.07	192	1.446	0.043	0.025	3.61
Other metabolites						
Taxifolin	26.33	179	1.804	0.035	0.011	31.784
Resorcinol	11.82	239	1.614	0.039	0.016	2.505
Alizarin	23.92	192	2.018	0.038	0.013	2.407

^1^ CON (control group) = dairy cows fed total mixed ration (TMR). ^2^ TR (treatment group) = dairy cows fed pelleted high-starch diet. ^3^ A metabolite is assigned as significantly different if its VIP is > 1 with and FDR < 0.05. ^4^ RT = retention time. ^5^ VIP = variable importance in projection. ^6^ FDR = false discovery rate. ^7^ FC = fold change, the ratio of mean value of peak area obtained from the cows in CON and TR groups. If FC > 1.5, means that this metabolite is higher in TR group than in CON group. If FC < 0.67, means that this metabolite is lower in TR group than in CON group.

**Table 8 metabolites-09-00154-t008:** Ingredient and chemical composition of CON ^1^ and TR ^2^ diets (DM ^3^ basis).

Item	CON	TR
Ingredient, %		
Barley	2.43	-
DDGS ^4^	2.21	-
Puffed soybean	2.81	-
Cottonseed	4.54	-
Beet pulp	3.70	-
Alfalfa hay	17.76	-
Corn silage	18.59	-
Fermented corn flour	5.05	-
Molasses	2.46	-
Oat hay	3.78	7.69
Maize	19.23	31.78
Vitamin and mineral mix ^5^	3.26	3.26
Soya bean meal	14.18	11.05
Corn germ meal	-	23.10
Corn husk	-	15.07
Soybean skin	-	8.05
Chemical composition		
Moisture, %	53.08	10.43
CP, %	16.96	16.78
NDF, %	28.17	32.33
ADF, %	15.09	15.19
Ether extract, %	4.28	4.19
Starch, %	16.52	21.59
NE_L_, Mcal/kg	1.57	1.63
NFC ^6^, %	42.47	38.35
Ash, %	8.12	8.35

^1^ CON (control group) = total mixed ration (TMR). ^2^ TR (treatment group) = pelleted high-starch diet. ^3^ DM = dry matter. ^4^ DDGS = distillers Dried Grains with Soluble. ^5^ Vitamin and mineral mix = formulated to contain 0.83% Ca; 0.34% P; 2.8% Mg; 3.5% K; 1500 mg/kg of Fe; 1500 mg/kg of Cu; 1495 mg/kg of Mn; 7000 mg/kg of Zn; 90 mg/kg of I; 20 mg/kg of Co; 50 mg/kg of Se; 500000 IU/kg of vitamin A; 130000 IU/kg of vitamin D; 1210 mg/kg of vitamin E. ^6^ NFC = non-fiber carbohydrate calculated by difference 100 − (% NDF + % CP + % Fat + % Ash).

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
