# Peer review of "The Ruminal Microbiome and Metabolome Alterations Associated with Diet-Induced Milk Fat Depression in Dairy Cows"

_metabolites, 2019, doi:10.3390/metabo9070154_

Round 1

Reviewer 1 Report

General

The authors have presented an interesting paper analysing the correlation between microbial populations and metabolites under a feeding regime that induces milk fat depression in dairy cows. MFD is an important problem in the dairy industry and this work starts to address this. However, I do have some concerns with the analysis (please see below).

The manuscript is generally well written with just a few minor grammatical errors e.g. “The ruminal microbiome provides many physiological functions that are lacking in the host of dairy cows” probably should be:  “The ruminal microbiome provides many physiological functions that are lacking in the host dairy cow.”

In table 3 the organic acids and short chain fatty acids (SCFA) are noted and in particular the microbial products (propionate, isobutyrate and butyrate were significantly different as has been observed before. These should also have been detected in the GCMS analysis. Can the authors confirm this? (They do not appear in Table 8) Being microbial in origin these metabolites should correlate to the metagenomic reads. Was this tested? If not this should be done.

Material and methods

Please describe how the cows were housed – were they allowed access to pasture (I assume not but this needs to be explicitly stated)

L 329 “5% orts” and L349 “feeds and orts”: please define- the term is not in general use

Statistical Analysis

Did the authors correct for multiple comparisons using the Bonferroni correction or equivalent? If so please note the number of features and the adjusted P Value. If not please do this. This should be done for the metagenomics and metabolomics data.

Conclusion

It cannot be said that the metabolites of the microbiota are necessarily changed as many of the metabolites may also be made by the animal or be present in food, especially amino acids, sugars etc the exception is some of the SCFA as noted above and it would be good to see if these correlate with any of the OTUs.

I recommend the manuscript be accepted after major revision if the analyses suggested above have not been carried out. If they have then the details need to be in the to improve the overall quality.

Author Response

Reviewer 1: The manuscript is generally well written with just a few minor grammatical errors e.g. “The ruminal microbiome provides many physiological functions that are lacking in the host of dairy cows” probably should be: “The ruminal microbiome provides many physiological functions that are lacking in the host dairy cow.” Response: We thank the reviewer for his valuable comments and we changed to “The ruminal microbiome provides many physiological functions that are lacking in the host dairy cow” in the revision. (Page 13, Line 360-361) In table 3 the organic acids and short chain fatty acids (SCFA) are noted and in particular the microbial products (propionate, isobutyrate and butyrate were significantly different as has been observed before. These should also have been detected in the GCMS analysis. Can the authors confirm this? (They do not appear in Table 8) Being microbial in origin these metabolites should correlate to the metagenomic reads. Was this tested? If not this should be done. Response: We thank the reviewer for his valuable comments. According to the reviewer’s suggestion, we rechecked the total metabolites again and didn’t find these VFAs. Maybe the aligned database of this company was not impeccable enough, so some metabolites could not be identified completely. But the measurement of VFAs in GC was very accurate, so we are sure the results of VFAs concentrations and percentages. We also added the correlation analysis between the VFA parameters and microbiota (Figure 3), and reword the sentence in my article in the revision. (Page 1, Line 36; Page 6, Line 175-193; Page 12, Line 309-326; Page 18, Line 557-560, 571) Material and methods Please describe how the cows were housed – were they allowed access to pasture (I assume not but this needs to be explicitly stated) Response: We thank the reviewer for his valuable comments and we stated as “Cows were housed in individual tie-stalls fitted with rubber mats and had continuous access to water from water bowls during the experiment” in the revision. (Page 14, Line 423-424) L 329 “5% orts” and L349 “feeds and orts”: please define- the term is not in general use orts are feed residual Response: We thank the reviewer for his valuable comments and we changed “5% orts” to “feed residual”, “feeds and orts” to “feeds and residual” in the revision. (Page 14, Line 431; Page 15, Line 451) Statistical Analysis Did the authors correct for multiple comparisons using the Bonferroni correction or equivalent? If so please note the number of features and the adjusted P Value. If not please do this. This should be done for the metagenomics and metabolomics data. Response: We thank the reviewer for his valuable comments and we used FDR (false discovery rate) to adjust the P value (Table 7), and we also reword the sentence in statistical analysis “DMs between both groups were identified combining VIP, FDR and FC (TR/CON) acquired from PLS-DA analysis and statistical analysis (VIP >1, FDR < 0.05 and FC > 1.5 or < 0.67)”. (Page 17, Line 550-552; Page 8, Table 7) Conclusion It cannot be said that the metabolites of the microbiota are necessarily changed as many of the metabolites may also be made by the animal or be present in food, especially amino acids, sugars etc the exception is some of the SCFA as noted above and it would be good to see if these correlate with any of the OTUs. Response: We thank the reviewer for his valuable comments and we deleted the words “of microbiota”. In the revision, we added the correlation analysis between the ruminal VFAs as noted and microbiota (Figure 3), and reword the words in my article. (Page 1, Line 36; Page 6, Line 175-193; Page 12, Line 309-326; Page 18, Line 557-560, 571)

Reviewer 2 Report

The authors have used a simple animal model to characterize the impact of high starch diets on the occurence MFD. The topic is relevant and original especially by combining diet induced changes in rumen microbiome and metabolome on a relevant syndrome in dairy cattle.

The following comments should be considered:

No data are given for energy intake in both groups.

Simply using starch in the TR group is an easy approach. A more detailed description on the starch source, its ruminal degradability, and the potential flow into the small intestines with subsequent enzymatic digestion would be necessary.

Taking ruminal samples via an OST. Such samples could be mixed with saliva samples with different proportions. How did the authors make sure that they have analyzed real rumen samples?

The authors did not differentiate between microbes present in the liquid phase and those which were particle associated. It has been shown by quite a number of studies that both compartments could be quite different.

Author Response

Reviewer 2: No data are given for energy intake in both groups. Response: We thank the reviewer for his valuable comments and we reanalyzed chemical composition of the diet, and we added the data in Table 8 (CON: NEL = 1.57 Mcal/kg of DM; TR: NEL = 1.63 Mcal/kg of DM) in the revision. (Page 14, Table 8) Simply using starch in the TR group is an easy approach. A more detailed description on the starch source, its ruminal degradability, and the potential flow into the small intestines with subsequent enzymatic digestion would be necessary. Response: We thank the reviewer for his valuable comments and we reword the sentence in discussion “TR diet had more proportion of maize with additional higher proportion of corn germ meal and corn husk, which provided large amounts of readily digestible carbohydrates. Mechanically activated and high temperature would change the physical structure of the starch and could enhance the starch degradability in the rumen (11). In the present study, the ingredients of the TR diet were chopped, ground and pelleted through high exit temperature (84℃) by the pelleted machine. This result indicated that TR diet could enhance the degradability of starch in the rumen and caused MFD in dairy cows better than CON diet” in the revision. (Page 11, Line 284-290) Taking ruminal samples via an OST. Such samples could be mixed with saliva samples with different proportions. How did the authors make sure that they have analyzed real rumen samples? Response: We thank the reviewer for his valuable comments about collecting ruminal samples. To reduce performance error among the cows, all samples were collected by 1 person, and every sampling site was controlled accurately. The same procedure was followed at each collection. Based on the former research, the inserting depth was 2 m in order to collect central rumen samples. When ruminal fluid samples were collected from each site, the first 150 mL was discharged into a 1,000-mL plastic beaker to avoid ruminal fluid mixed with saliva samples and then 40 ml was collected. After sampling each cow, the ruminal fluid in the plastic beaker was turned back into the rumen and the OST was washed using running water. The authors did not differentiate between microbes present in the liquid phase and those which were particle associated. It has been shown by quite a number of studies that both compartments could be quite different. Response: We thank the reviewer for his valuable comments and we used OST (the method described by Shen (2012) Journal of Dairy Science) to collect rumen samples. From the diameter of the holes (2.5 mm), the samples we collected were rumen content, and then we put them in the syringe and strained through 4 layers of cheesecloth. So the final rumen samples for analysis were ruminal fluid instead of rumen content associated with particles.

Reviewer 3 Report

General comments:

This manuscript associates the effects of milk fat depression induced by diet with ruminal microbiome and metabolome. The topic is of interest to both academic and industry audiences because it describes ruminal microbiomes, metabolites, and its potential associations with milk quality traits. However, the manuscript has a number of major limitations in the writing, experimental design and interpretation of results.

Abstract

Line 16: lactation stage define

Line 17 : delete the word “pelleted”.  Already described as TMR

Line 18: reworded the sentence. Results are the outcome of the experiment, not showing

Line 19: increase the abundance in which tissue?

Needs a conclusion. Which mechanisms are potentially involved in MFD

Introduction

The overall introduction is outdated and disconnected from the objectives

Line 32 to 37: reword the sentences. The interaction diet, microbiome and tissue physiology is a classic interaction a theory or a believe? Please clarify

Line 37: update the reference (#3)

Line 38: ratio acetate: propionate?Explain why VFA reduce milk fat synthesis

Line 39 to 41: reworded the sentence. Link VFA with changes in the microbiome

Line 41-42/43-44: instead of report others authors results, described the importance of previous findings.

Line: 44 delete yet

Line 45: why is important the information on the metabolites of ruminal microbiota?

Line 46: which evidence?

Line 50: image?

Line 48-51: revise the proper use of English

Line 52: delete “there would be” revise English grammar

MATERIALS AND METHODS

Why the feeding experiment is 55 days?

Line 319: DMI=dry matter intake? unit

Line 319: multiparous define (xx±xx parities) report values on average±sd

Line 319: describe week of lactation of the cows

Line 407: Standarize the use of the unit ml or µl

Line 408: samples were kept for how long?

Line 411: BSTFA-abbreviation was described before? Trifluoroacetamide??

Line 412: Pegasus HT (Company, city, state, country) Be consistent through the manuscript

Line 413: Agilent 7890, software?

Line 426/428: Multivariate/Variable change it to minuscule letters

Line 429: statitical test? T-test?

Line:432 intensity, not concentration

Which quality control was used for data quality?

How metabolomics data was process?

Which software?

Which database was used for metabolite identification?

Line 434: Gephi 0.8.2 citation

Results:

Delete throughout the results section “those in”

Adjust the p-values number next to the effects.

Interpret the results obtained, don’t repeat the table.

Line 62: delete (P > 0.05)

Line 92: how differ? The sentence is meaningless

Lin 149: change obtain to identified. How did you identify the metabolites? Describe

Line 151 to 152: further analysis? Describe

Line 154: obvious?

Line 201: From above results?,  we can also find that? Delete

Line 184: using which software?

In table 8 is not possible to identify which metabolite is higher or lower in the con vs tr groups

Line 187: enriched in TR group compared with CON group. How? Described

Discussion:

Line 213; easily? How?

Line 216: various feeding regimens? Not related to the experiment

Update all your references. Sutton is a review from 1989.

Line 217: can be?

Line 220: possible reason?          How? Describe

Line 226: lot of researchers??

Line 230: It is not surprising? Use appropriate scientific English description

Line 221: feature?

Line 231 to 233: report particles sizes of the diets control vs tr. Both are TMR, differences come from the composition of the diets, not in particles size

Line 240: follow the guidelines of the journal to use appropriate references system throughout the manuscript

Line 257: intensively concerned?

Line 261: to be sensitive?

Paragraph 234 to 266 is not a discussion. Only repeat results and compare with other values reported.

Line 267-270: The sentence has no sense. Saleem didn’t observe that.

Overall discussion and conclusion need a major revision. What are the implications?

Author Response

Reviewer 3: Abstract Line 16: lactation stage define Response: We thank the reviewer for his valuable comment and we added the Days in Milk in the revision. (Page 1, Line 28) Line 17: delete the word “pelleted”. Already described as TMR Response: We thank the reviewer for his valuable comment and we deleted the word “pelleted” in the revision. (Page 1, Line 29) Line 18: reworded the sentence. Results are the outcome of the experiment, not showing Response: We thank the reviewer for his valuable comment and we reword the sentences in the revision. (Page 1, Line 31-32) Line 19: increase the abundance in which tissue? Response: We thank the reviewer for his valuable comment and we stated that it is in the ruminal fluid in the revision. (Page 1, Line 34) Needs a conclusion. Which mechanisms are potentially involved in MFD. Response: We thank the reviewer for his valuable comment and we changed to “In general, these findings revealed that TR diet reduced the rumination time and altered rumen fermentation type, which lead to changes in the composition of ruminal bacteria and metabolites, and caused MFD” in the revision. (Page 1, Line 39-41) Introduction Line 32 to 37: reword the sentences. The interaction diet, microbiome and tissue physiology is a classic interaction a theory or a believe? Please clarify Response: We thank the reviewer for his valuable comment and we changed to “There have been various proposed theories to explain the occurrence of MFD in dairy industry, and most widely accepted theory was that under the condition of large-scale dairy farm, dairy cows fed high concentrate diets had a decrease of milk fat because of the changes in the concentration of volatile fatty acids (VFA) and the alteration of microbiota (3) in the rumen” in the revision. (Page 2, Line 48-52) Line 37: update the reference (#3) Response: We thank the reviewer for his valuable comment and we update the reference in the revision. (Page 2, Line 52) Line 38: ratio acetate: propionate? Explain why VFA reduce milk fat synthesis Response: We thank the reviewer for his valuable comment and we rewrote this expression as “acetate: propionate”. In ruminants, about one-half of the milk fatty acids (molar percent) are derived from de novo synthesis. Ruminants utilize acetate produced in rumen fermentation of carbohydrates as the major carbon source. In addition, β-hydroxybutyrate, produced by the rumen epithelium from absorbed butyrate, provides about one half of the first four carbons of de novo synthesized fatty acids in the ruminant. In our study, the acetate and butyrate percentage in rumen were reduced in TR group compared with CON group, so the decreased carbon source can explain why the milk fat de novo synthesis was reduced. (Page 2, Line 53) Line 39 to 41: reworded the sentence. Link VFA with changes in the microbiome Response: We thank the reviewer for his valuable comments and we reword the sentence “Regarding the relationship between the ruminal microbiota and MFD” to “The accumulation of VFA is closely related to microbial fluctuations, thus the changes in VFA in the case of MFD indicated an association between the ruminal microbiota and MFD” in the revision. (Page 2, Line 54-56) Line 41-42/43-44: instead of report others authors results, described the importance of previous findings. Response: We thank the reviewer for his valuable comment and we want to elaborate the ruminal bacterial community composition can be altered in MFD cattle. And these reports did not focus attention on metabolic changes of ruminal microbiota, so the aim of our study was to find some associations between microbiome and metabolome in MFD cattle. We also reword the sentence in the introduction in the revision. (Page 2, Line 62-68) Line: 44 delete yet Response: We thank the reviewer for his valuable comments and we deleted the word “yet”. (Page 2, Line 62) Line 45: why is important the information on the metabolites of ruminal microbiota? Response: We thank the reviewer for his valuable comments and since most of the metabolites used by the animal body for growth and production are synthesized in the rumen by the rumen microorganisms, it would be very beneficial to discover the relationship between the rumen microorganisms and the metabolites found in the ruminal fluid. Line 46: which evidence? Response: We thank the reviewer for his valuable comment and references are cited in the end of the sentence in the revision. (Page 19, Line 600-602) Line 50: image? Response: We thank the reviewer for his valuable comment and we amended this phrase to be “Furthermore, it can provide comprehensive associations between the ruminal microbiome and metabolome under conditions of diet-induced MFD” in the revision. (Page 2, Line 70-72) Line 48-51: revise the proper use of English Response: We thank the reviewer for his valuable comment and we reword the sentence in the revision. (Page 2, Line 73-76) Line 52: delete “there would be” revise English grammar Response: We thank the reviewer for his comment and we delete “there would be”, and reword the sentence in the revision. (Page 2, Line 73) MATERIALS AND METHODS Why the feeding experiment is 55 days? Response: We thank the reviewer for the comments and we found that 55 days was adequate for us to detect the significantly decreased milk fat percentage in TR group when compared with the CON group. Line 319: DMI=dry matter intake? unit Response: We thank the reviewer for his valuable comment and DIM is Days in Milk, not Dry Matter Intake. We defined the DIM at the first place it occurred in the manuscript. Page 14, Line 420) Line 319: multiparous define (xx ± xx parities) report values on average ± sd Response: We thank the reviewer for the comments and all the cows were second parity during the experiment and we reword the sentence in the revision. (Page 1, Line 27; Page 14, Line 419-420) Line 319: describe week of lactation of the cows Response: We thank the reviewer for his valuable comment and DIM was mentioned in the revision. (Page 14, Line 420) Line 407: Standardize the use of the unit ml or µl Response: We thank the reviewer for his valuable comment and we standardized the use of µl in the revision. (Page 17, Line 525-526) Line 408: samples were kept for how long? Response: We thank the reviewer for his valuable comment and samples were kept about 2 days and then for analysis in the revision. Line 411: BSTFA-abbreviation was described before? Trifluoroacetamide?? Response: We thank the reviewer for his valuable comment and BSTFA: Chlorotrimethylsilane Silylating Reagent and we changed to “Chlorotrimethylsilane Silylating Reagent” in the revision. (Page 17, Line 529-530) Line 412: Pegasus HT (Company, city, state, country) be consistent through the manuscript Response: We thank the reviewer for his valuable comment and we changed it to “Pegasus HT (LECO, Shanghai, China)” in the revision. (Page 17, Line 532) Line 413: Agilent 7890, software? Response: We thank the reviewer for his valuable comment and Agilent 7890 is the model of gas chromatograph machine in the revision. (Page 17, Line 531) Line 426/428: Multivariate/Variable change it to minuscule letters Response: We thank the reviewer for his valuable comment and we reword the sentence in the revision. (Page 17, Line 550-551) Line 429: statitical test? T-test? Response: We thank the reviewer for his valuable comment and P value of each metabolite between two groups was calculated using t-test, which was subsequently correlated to obtain FDR. Differential metabolites between two groups were identified by combining variable VIP, FDR, and FC (TR/CON) with the criteria of VIP >1, FDR < 0.05, and FC > 1.5 or < 0.67. (Page 17, Line 550-552) Line: 432 intensity, not concentration Response: We thank the reviewer for his valuable comment and we deleted this word because we used peak area for analysis. (Page 18, Line 555) Which quality control was used for data quality? Response: We thank the reviewer for his valuable comment and internal standard and Methyl saturated fatty acids were used for data quality. It is necessary to monitor the stability and signal of the instrument in real time during the detection process, find the abnormality in time, and eliminate the problem as soon as possible to ensure the quality of the final data collected. How metabolomics data was process? Response: We thank the reviewer for his valuable comment and P value of each metabolite between two groups was calculated using t-test, which was subsequently correlated to obtain FDR. Differential metabolites between two groups were identified by combining VIP, FDR, and FC (TR/CON) with the criteria of VIP >1, FDR < 0.05, and FC > 1.5 or < 0.67. (Page 17, Line 550-552) Which software? Response: We thank the reviewer for his valuable comment and we used the SIMCA-P+ 13.0 software (Umea, Sweden) to analyze the differential metabolites, we have listed it in the revision. (Page 17, Line 548) Which database was used for metabolite identification? Response: We thank the reviewer for his valuable comment and LECO-Fiehn Rtx-5 database (Shanghai Biotreee biotech Co. Ltd.) was utilized to identify metabolites. Line 434: Gephi 0.8.2 citation Response: We thank the reviewer for his valuable comment and Gephi 0.8.2 is a kind of software for analyzing the correlation between microbiota and metabolites and we changed it to “Gephi 0.8.2 beta software (https://gephi.org/)” in the revision. (Page 18, Line 562) Results: Delete throughout the results section “those in” Response: We thank the reviewer for his valuable comment and we deleted this phrase throughout all the results section in the revision. (Page 2-11) Adjust the p-values number next to the effects. Response: We thank the reviewer for his valuable comment and we adjust all the P-value next to the effects in the revision. (Page 2-11) Interpret the results obtained, don’t repeat the table. Response: We thank the reviewer for his valuable comment and we resembled the sentence in the results in the revision. (Page 2-11) Line 62: delete (P > 0.05) Response: We thank the reviewer for his valuable comment and we deleted it in the revision. (Page 2) Line 92: how differ? The sentence is meaningless Response: We thank the reviewer for his valuable comments and we deleted the sentence in the revision. (Page 4, Line 114-115) Lin 149: change obtain to identified. How did you identify the metabolites? Describe Response: We thank the reviewer for his valuable comment and we replace “obtained” by “identified”. Chroma TOF 4.3X software of LECO Corporation and LECO-Fiehn Rtx-5 database were used for raw peaks exacting, the data baselines filtering and calibration of the baseline, peak alignment, deconvolution analysis, peak identification and integration of the peak area. Both of mass spectrum match and retention index match were considered in metabolites identification. (Page 7, Line 195) Line 151 to 152: further analysis? Describe Response: We thank the reviewer for his valuable comment and we deleted this phrase in the revision. (Page 7, Line 199) Line 154: obvious? Response: We thank the reviewer for his valuable comment and we rewrote the sentence to be “The ruminal metabolites in CON and TR groups were clearly distinct, demonstrating that the difference of metabolites in the rumen between CON and TR groups.” (Page 7, Line 200-203) Line 201: From above results? we can also find that? Delete Response: We thank the reviewer for his valuable comment and we deleted these words in the revision. (Page 10, Line 253-254) Line 184: using which software? Response: Differential metabolite data were used for pathway enrichment analysis on the MetaboAnalyst 3.0 (http://www.metaboanalyst.ca). In table 8 is not possible to identify which metabolite is higher or lower in the con vs tr groups Response: We thank the reviewer for his valuable comment and all these metabolites in table 7 (changed) are different metabolites, which means that their VIP >1, FDR < 0.05, and FC > 1.5 or < 0.67. FC > 1.5 means differential metabolites were higher in TR group than CON group, while FC < 0.67 means differential metabolites were lower in TR group than CON group. (Page 17, Line 550-552) Line 187: enriched in TR group compared with CON group. How? Described Response: We thank the reviewer for his valuable. We rewrote this sentence as reading "Results revealed that glycine, serine and threonine metabolism, biosynthesis of amino-acetyl-tRNA, methane metabolism; amino acids and nucleoside metabolism, arginine and alanine metabolism, glutamate metabolism were significantly enriched (P < 0.05) by differential metabolites between the CON and TR groups." in the revision (Page 9, Line 235-239) Discussion: Line 213 easily? How? Response: We thank the reviewer for his valuable comment and fat percentage in milk is related by fiber digestion in the rumen which is considered a very sensitive process to the rumen environment optimization level. Line 216: various feeding regimens? Not related to the experiment Response: We thank the reviewer for his valuable comment and we just need to highlight the relationship between VFA patterns and milk fat level. Update all your references. Sutton is a review from 1989. Response: We thank the reviewer for his valuable comment and we update all the references in the revision. (Page 11, Line 268) Line 217: can be? Response: We thank the reviewer for his valuable comment and we reword the sentence in the revision. (Page 11, Line 269) Line 220: possible reason? How? Describe Response: We thank the reviewer for his valuable comment and we reword the sentence. From the former research found that the decrease in the concentration of acetate and butyrate in MFD cattle, but in our study the percentage of acetate and butyrate were significantly decreased in TR group compared with CON group, and we believed that the possible reason might be the decrease in percentage of acetate and butyrate in MFD cattle. Line 226: lot of researchers?? Response: We thank the reviewer for his valuable comment and we changed to “various studies” in the revision. (Page 11, Line 279) Line 230: It is not surprising? Use appropriate scientific English description Response: We thank the reviewer for his valuable comment and we used “expectedly” in the revision. (Page 12, Line 290) Line 221: feature? Response: We thank the reviewer for this comment and we used the word “characteristics” in the revision. (Page 11, Line 274) Line 231 to 233: report particles sizes of the diets control vs tr. Both are TMR, differences come from the composition of the diets, not in particles size Response: We thank the reviewer for this comment and we agree with it. Both the composition of diets and the sizes of the diets could affect milk fat percentage. Because diet composition alters metabolism directly while the particle size of diet has an effect on rumination time. In this manuscript, we just focus on the effect of pellet high-starch diet on ruminal microbiome, metabolome, and milk fat, but don’t pay much attention on which factor is the dominant. Line 240: follow the guidelines of the journal to use appropriate references system throughout the manuscript Response: We thank the reviewer for his valuable comment and we reword the sentence in the revision. Line 257: intensively concerned? Response: We thank the reviewer for his valuable comment and we reword the sentence in the revision. (Page 11-12, Line 294-326) Line 261: to be sensitive? Response: We thank the reviewer for his valuable comment and we reword the sentence in the revision. (Page 11-12, Line 294-326) Paragraph 234 to 266 is not a discussion. Only repeat results and compare with other values reported. Response: We thank the reviewer for his valuable comment and we reword the sentence in the revision. (Page 11-12, Line 294-326) Line 267-270: The sentence has no sense. Saleem didn’t observe that. Response: We thank the reviewer for his valuable comment and we reword the sentence in the revision. (Page 11-12, Line 294-326) Overall discussion and conclusion need a major revision. What are the implications? Response: We thank the reviewer for his valuable and we resemble the words in discussion and conclusion in the revision.

Round 2

Reviewer 1 Report

The authors have improved their paper. I believe the P values in the text should not be deleted however, and there are some minor English language changes needed. e.g. I'm not sure what the authors mean in line 670 where they say "..acids twisted from linoleic." The use of twisted is clearly incorrect.

Author Response

The authors have improved their paper. I believe the P values in the text should not be deleted however, and there are some minor English language changes needed. e.g. I'm not sure what the authors mean in line 670 where they say "..acids twisted from linoleic." The use of twisted is clearly incorrect.

Response: We thank the reviewer for his valuable comments and we added the P value in Table 7. And we changed back to “The dicarboxylic acids were produced by the bio-hydrogenation of the non-conjugated linoleic acids formed from linoleic [29, 30].” (Page 13, Line 388)

Reviewer 2 Report

The comments have been considered adequately!

Author Response

We thanks the reviewer!

Reviewer 3 Report

I appreciate all the changes performed in the manuscript but the lines number described in the comments do not correspond in the manuscript. In addition, line numbers are not continuous throughout the manuscript.  Please revise

Author Response

I appreciate all the changes performed in the manuscript but the lines number described in the comments do not correspond in the manuscript. In addition, line numbers are not continuous throughout the manuscript. Please revise

Response: We thank the reviewer for his valuable comments and we revised the line numbers throughout my manuscript.

Round 3

Reviewer 3 Report

Changes in the manuscript are not possible to track. Line numbers do not correspond between manuscript and authors response. Changes perform in the manuscript must correspond with the description in your comments. Please provide a new version of your response (1) with the correct corresponding number of lines of the changes performed.

Author Response

We thank the reviewer for his valuable comments and we rechecked the Line number.

Abstract

Line 16: lactation stage define

Response: We thank the reviewer for his valuable comment and we added the Days in Milk in the revision. (Page 1, Line 22)

Line 17: delete the word “pelleted”. Already described as TMR

Response: We thank the reviewer for his valuable comment and we deleted the word “pelleted” in the revision. (Page 1, Line 23)

Line 18: reworded the sentence. Results are the outcome of the experiment, not showing

Response: We thank the reviewer for his valuable comment and we reword the sentences in the revision. (Page 1, Line 24-25)

Line 19: increase the abundance in which tissue?

Response: We thank the reviewer for his valuable comment and we stated that it is in ruminal fluid in the revision. (Page 1, Line 27)

Needs a conclusion. Which mechanisms are potentially involved in MFD.

Response: We thank the reviewer for his valuable comment and we changed to “In general, these findings revealed that TR diet reduced the rumination time and altered rumen fermentation type, which lead to changes in the composition of ruminal microbiota and metabolites, and caused MFD” in the revision. (Page 1, Line 31-33)

Introduction

Line 32 to 37: reword the sentences. The interaction diet, microbiome and tissue physiology is a classic interaction a theory or a believe? Please clarify

Response: We thank the reviewer for his valuable comment and we changed to “There have been various proposed theories to explain the occurrence of MFD in dairy industry, and most widely accepted theory was that under the condition of large-scale dairy farm, dairy cows fed high concentrate diets had a decrease of milk fat because of the changes in the concentration of volatile fatty acids (VFA) and the alteration of microbiota in the rumen (3)” in the revision. (Page 1, Line 39-43)

Line 37: update the reference (#3)

Response: We thank the reviewer for his valuable comment and we update the reference in the revision. (Page 1, Line 43)

Line 38: ratio acetate: propionate? Explain why VFA reduce milk fat synthesis

Response: We thank the reviewer for his valuable comment and we rewrote this expression as “acetate: propionate”. In ruminants, about one-half of the milk fatty acids (molar percent) are derived from de novo synthesis. Ruminants utilize acetate produced in rumen fermentation of carbohydrates as the major carbon source. In addition, β-hydroxybutyrate, produced by the rumen epithelium from absorbed butyrate, provides about one half of the first four carbons of de novo synthesized fatty acids in the ruminant. In our study, the acetate and butyrate percentage in rumen were reduced in TR group compared with CON group, so the decreased carbon source can explain why the milk fat de novo synthesis was reduced. (Page 1, Line 44)

Line 39 to 41: reworded the sentence. Link VFA with changes in the microbiome

Response: We thank the reviewer for his valuable comments and we reword the sentence “Regarding the relationship between the ruminal microbiota and MFD” to “The accumulation of VFA is closely related to microbial fluctuations, thus the changes in VFA in the case of MFD indicated an association between the ruminal microbiota and MFD” in the revision. (Page 2, Line 45-46)

Line 41-42/43-44: instead of report others authors results, described the importance of previous findings.

Response: We thank the reviewer for his valuable comment and we want to elaborate the ruminal bacterial community composition can be altered in MFD cattle. And these reports did not focus attention on metabolic changes of ruminal microbiota, so the aim of our study was to find some associations between microbiome and metabolome in MFD cattle. We also reword the sentence in the introduction in the revision. (Page 2, Line 51-53)

Line: 44 delete yet

Response: We thank the reviewer for his valuable comments and we deleted the word “yet”. (Page 2, Line 52)

Line 45: why is important the information on the metabolites of ruminal microbiota?

Response: We thank the reviewer for his valuable comments and since most of the metabolites used by the animal body for growth and production are produced by the rumen microorganisms, it would be very beneficial to discover the relationship between the rumen microorganisms and the metabolites found in the ruminal fluid.

Line 46: which evidence?

Response: We thank the reviewer for his valuable comment and references are cited in the end of the sentence in the revision. (Page 16, Line 473-473)

Line 50: image?

Response: We thank the reviewer for his valuable comment and we amended this phrase to be “Furthermore, it can provide comprehensive associations between the ruminal microbiome and metabolome under conditions of diet-induced MFD” in the revision. (Page 2, Line 60-62)

Line 48-51: revise the proper use of English

Response: We thank the reviewer for his valuable comment and we reword the sentence in the revision. (Page 2, Line 59-62)

Line 52: delete “there would be” revise English grammar

Response: We thank the reviewer for his comment and we delete “there would be”, and reword the sentence in the revision. (Page 2, Line 59)

MATERIALS AND METHODS

Why the feeding experiment is 55 days?

Response: We thank the reviewer for the comments and we found that 55 days was adequate for us to detect the significantly decreased milk fat percentage in TR group when compared with the CON group.

Line 319: DMI=dry matter intake? unit

Response: We thank the reviewer for his valuable comment and DIM is Days in Milk, not Dry Matter Intake. We defined the DIM at the first place it occurred in the manuscript. (Page 12, Line 329)

Line 319: multiparous define (xx ± xx parities) report values on average ± sd

Response: We thank the reviewer for the comments and all the cows were second parity during the experiment and we reword the sentence in the revision. (Page 1, Line 21; Page 12, Line 328-329)

Line 319: describe week of lactation of the cows

Response: We thank the reviewer for his valuable comment and DIM was mentioned in the revision. (Page 12, Line 329)

Line 407: Standardize the use of the unit ml or µl

Response: We thank the reviewer for his valuable comment and we standardized the use of µl in the revision (Page 15, Line 410-411).

Line 408: samples were kept for how long?

Response: We are very sorry for this mistake, and we have rephrased it as reading “Samples were centrifuged at 12,000 rpm for 15 min at 4” in the revision. (Page 15, Line 409-410)

Line 411: BSTFA-abbreviation was described before? Trifluoroacetamide??

Response: We thank the reviewer for his valuable comment. We have rephrased this sentence as reading “70 μL of chlorotrimethylsilane silylating reagent was added into the aliquots after 1.5 h incubation at 70.” in the revision. (Page 15, Line 413)

Line 412: Pegasus HT (Company, city, state, country) be consistent through the manuscript

Response: We thank the reviewer for his valuable comment and we changed it to “Pegasus HT (LECO, Shanghai, China)” in the revision. (Page 15, Line 414)

Line 413: Agilent 7890, software?

Response: We thank the reviewer for his valuable comment and Agilent 7890 is the model of gas chromatograph machine in the revision. (Page 15, Line 414)

Line 426/428: Multivariate/Variable change it to minuscule letters

Response: We thank the reviewer for his valuable comment and we reword the sentence in the revision. (Page 15, Line 428-429)

Line 429: statistical test? T-test?

Response: We thank the reviewer for his valuable comment and P value of each metabolite between two groups was calculated using t-test, which was subsequently correlated to obtain FDR. Differential metabolites between two groups were identified by combining variable VIP, FDR, and FC (TR/CON) with the criteria of VIP >1, FDR < 0.05, and FC > 1.5 or < 0.67. (Page 15, Line 429-431)

Line: 432 intensity, not concentration

Response: We thank the reviewer for his valuable comment and we deleted this word because we used peak area for analysis. (Page 15, Line 434)

Which quality control was used for data quality?

Response: We thank the reviewer for his valuable comment. In the present study, to ensure data quality, quality control samples were inserted randomly. We have added this information in the revision. (Page 15, Line 431)

How metabolomics data was process?

Response: We thank the reviewer for his valuable comment and P value of each metabolite between two groups was calculated using t-test, which was subsequently correlated to obtain FDR. Differential metabolites between two groups were identified by combining VIP, FDR, and FC (TR/CON) with the criteria of VIP >1, FDR < 0.05, and FC > 1.5 or < 0.67. (Page 15, Line 429-431)

Which software?

Response: We thank the reviewer for his valuable comment and we used the SIMCA-P+ 13.0 software (Umea, Sweden) to analyze the differential metabolites, we have listed it in the revision. (Page 15, Line 428)

Which database was used for metabolite identification?

Response: We thank the reviewer for his valuable comment and LECO-Fiehn Rtx-5 database (Shanghai Biotreee biotech Co. Ltd.) was utilized to identify metabolite. (Page 15, Line 420-421)

Line 434: Gephi 0.8.2 citation

Response: We thank the reviewer for his valuable comment and Gephi 0.8.2 is a kind of software for analyzing the correlation between microbiota and metabolites and we changed it to “Gephi 0.8.2 beta software (https://gephi.org/)” in the revision. (Page 15, Line 440)

Results:

Delete throughout the results section “those in”

Response: We thank the reviewer for his valuable comment and we deleted this phrase throughout all the results section in the revision. (Page 2-10)

Adjust the p-values number next to the effects.

Response: We thank the reviewer for his valuable comment and we adjust all the P-value next to the effects in the revision. (Page 2-10)

Interpret the results obtained, don’t repeat the table.

Response: We thank the reviewer for his valuable comment and we resembled the sentence in the results in the revision. (Page 2-10)

Line 62: delete (P > 0.05)

Response: We thank the reviewer for his valuable comment and we deleted it in the revision. (Page 2, Line 68; Line 76)

Line 92: how differ? The sentence is meaningless

Response: We thank the reviewer for his valuable comments and we deleted the sentence in the revision. (Page 3, Line 93)

Lin 149: change obtain to identified. How did you identify the metabolites? Describe

Response: We thank the reviewer for his valuable comment and we replace “obtained” by “identified”. Chroma TOF 4.3X software of LECO Corporation and LECO-Fiehn Rtx-5 database were used for raw peaks exacting, the data baselines filtering and calibration of the baseline, peak alignment, deconvolution analysis, peak identification and integration of the peak area. Both of mass spectrum match and retention index match were considered in metabolites identification. (Page 7, Line 159)

Line 151 to 152: further analysis? Describe

Response: We thank the reviewer for his valuable comment and we deleted this phrase in the revision. (Page 7, Line 162)

Line 154: obvious?

Response: We thank the reviewer for his valuable comment and we rewrote the sentence to be “The ruminal metabolites in CON and TR groups were clearly distinct, demonstrating that the difference of metabolites in the rumen between CON and TR groups.” (Page 7, Line 164-165)

Line 201: From above results? we can also find that? Delete

Response: We thank the reviewer for his valuable comment and we deleted these words in the revision. (Page 10, Line 212)

Line 184: using which software?

Response: Differential metabolite data were used for pathway enrichment analysis on the MetaboAnalyst 3.0 (http://www.metaboanalyst.ca).

In table 8 is not possible to identify which metabolite is higher or lower in the con vs tr groups

Response: We thank the reviewer for his valuable comment and all these metabolites in table 7 (changed) are different metabolites, which means that their VIP >1, FDR < 0.05, and FC > 1.5 or < 0.67. FC > 1.5 means differential metabolites were higher in TR group than CON group, while FC < 0.67 means differential metabolites were lower in TR group than CON group. (Page 15, Line 429-431)

Line 187: enriched in TR group compared with CON group. How? Described

Response: We thank the reviewer for his valuable. We rewrote this sentence as reading "Results revealed that glycine, serine and threonine metabolism, biosynthesis of amino-acetyl-tRNA, methane metabolism; amino acids and nucleoside metabolism, arginine and alanine metabolism, glutamate metabolism were significantly enriched (FDR < 0.05) by differential metabolites between the CON and TR groups." in the revision (Page 9, Line 193-196)

Discussion:

Line 213 easily? How?

Response: We thank the reviewer for his valuable comment. We deleted the word “easily” in the revision. (Page 10, Line 221)

Line 216: various feeding regimens? Not related to the experiment

Response: We thank the reviewer for his valuable comment and we have deleted this section in the revision. (Page 10, Line 223)

Update all your references. Sutton is a review from 1989.

Response: We thank the reviewer for his valuable comment and we update all the references in the revision. (Page 10, Line 223-224)

Line 217: can be?

Response: We thank the reviewer for his valuable comment and we reword the sentence in the revision. (Page 10, Line 224)

Line 220: possible reason? How? Describe

Response: We thank the reviewer for his valuable comment and we reword the sentence. Previous studies found that the decrease in the concentration of acetate and butyrate in MFD cattle, but in our study, the percentage of acetate and butyrate were significantly decreased in TR group compared with CON group, and we believed that the possible reason might be the decrease in percentage of acetate and butyrate in MFD cattle.

Line 226: lot of researchers??

Response: We thank the reviewer for his valuable comment and we changed to “various studies” in the revision. (Page 10, Line 234)

Line 230: It is not surprising? Use appropriate scientific English description

Response: We thank the reviewer for his valuable comment and we used “expectedly” in the revision. (Page 11, Line 244)

Line 221: feature?

Response: We thank the reviewer for this comment and we used the word “characteristics” in the revision. (Page 10, Line 229)

Line 231 to 233: report particles sizes of the diets control vs tr. Both are TMR, differences come from the composition of the diets, not in particles size

Response: We thank the reviewer for this comment and we agree with it. Both the composition of diets and the sizes of the diets could affect milk fat percentage. Because diet composition alters metabolism directly while the particle size of diet has an effect on rumination time. In this manuscript, we just focus on the effect of pellet high-starch diet on ruminal microbiome, metabolome, and milk fat, but don’t pay much attention on which factor is the dominant.

Line 240: follow the guidelines of the journal to use appropriate references system throughout the manuscript

Response: We thank the reviewer for his valuable comment and we have checked all references in the revision.

Line 257: intensively concerned?

Response: We thank the reviewer for his valuable comment and we rephrased this section in the revision. (Page 11, Line 248-287)

Line 261: to be sensitive?

Response: We thank the reviewer for his valuable comment and we reword the section in the revision. (Page 11, Line 248-287)

Paragraph 234 to 266 is not a discussion. Only repeat results and compare with other values reported.

Response: We thank the reviewer for his valuable comment and we reword the sentence in the revision. (Page 11, Line 248-287)

Line 267-270: The sentence has no sense. Saleem didn’t observe that.

Response: We thank the reviewer for his valuable comment and we reword the sentence in the revision. (Page 11, Line 248-287)

Overall discussion and conclusion need a major revision. What are the implications?

Response: We thank the reviewer for his valuable and we rewrote the discussion and conclusion in the revision.

Round 4

Reviewer 3 Report

Line 51: elaborated? Might be proposed?

Line 56: delete” help to”

Line 421: describe metabolomics data processing, e.g. alignment, normalization, peak detection, deconvolution, homogenization

Line 251/255: Alejandro?

Line 281: lacking??

Author Response

Line 51: elaborated? Might be proposed?

We thank the reviewer for his valuable comments and we changed to “proposed”. (Page 2, Line 51)

Line 56: delete” help to”

We thank the reviewer for his valuable comments and we deleted “help to”. (Page 2, Line 56)

Line 421: describe metabolomics data processing, e.g. alignment, normalization, peak detection, deconvolution, homogenization

We thank the reviewer for his valuable comments and we reworded the sentence “Chroma TOF 4.3X software and LECO-Fiehn Rtx5 database (Shanghai Biotreee biotech Co. Ltd.) were used for raw peaks exacting, the data baselines filtering and calibration of the baseline, peak alignment, deconvolution analysis, peak identification and integration of the peak area. LECO-Fiehn Rtx-5 database was utilized to identify metabolites”. (Page 15, Line 420-423)

Line 251/255: Alejandro?

We thank the reviewer for his valuable comments and we changed to “Belanche et al.”. (Page 11, Line 251/255)

Line 281: lacking??

We thank the reviewer for his valuable comments and we changed to “The ruminal microbiome provides many physiological functions that are required by the host dairy cow”. (Page 11, Line 280-281)